

# A probabilistic model to recover individual genomes from metagenomes

Johannes Dröge[1], Alexander Schönhuth[2] and Alice C. McHardy[1]

[1] Computational Biology of Infection Research, Helmholtz Centre for Infection Research, Braunschweig, Germany
[2] Centrum Wiskunde & Informatica, Amsterdam, The Netherlands

## ABSTRACT

Shotgun metagenomics of microbial communities reveal information about strains of relevance for applications in medicine, biotechnology and ecology. Recovering their genomes is a crucial but very challenging step due to the complexity of the underlying biological system and technical factors. Microbial communities are heterogeneous, with oftentimes hundreds of present genomes deriving from different species or strains, all at varying abundances and with different degrees of similarity to each other and reference data. We present a versatile probabilistic model for genome recovery and analysis, which aggregates three types of information that are commonly used for genome recovery from metagenomes. As potential applications we showcase metagenome contig classification, genome sample enrichment and genome bin comparisons. The open source implementation MGLEX is available via the *Python Package Index* and on *GitHub* and can be embedded into metagenome analysis workflows and programs.

## INTRODUCTION

Shotgun sequencing of DNA extracted from a microbial community recovers genomic data from different community members while bypassing the need to obtain pure isolate cultures. It thus enables novel insights into ecosystems, especially for those genomes which are inaccessible by cultivation techniques and isolate sequencing. However, current metagenome assemblies are oftentimes highly fragmented, including unassembled reads, and require further processing to separate data according to the underlying genomes. Assembled sequences, called contigs, that originate from the same genome are placed together in this process, which is known as metagenome binning (*Tyson et al., 2004*; *Dröge & McHardy, 2012*) and for which many programs have been developed. Some are trained on reference sequences, using contig *k*-mer frequencies or sequence similarities as sources of information (*McHardy et al., 2007*; *Dröge, Gregor & McHardy, 2014*; *Wood & Salzberg, 2014*; *Gregor et al., 2016*), which can be adapted to specific ecosystems. Others cluster the contigs into genome bins, using contig *k*-mer frequencies and read coverage (*Chatterji et al., 2008*; *Kislyuk et al., 2009*; *Wu et al., 2014*; *Nielsen et al., 2014*; *Imelfort et al., 2014*; *Alneberg et al., 2014*; *Kang et al., 2015*; *Lu et al., 2017*).

Recently, oftentimes multiple biological or technical samples of the same environment are sequenced to produce distinct genome copy numbers across samples, sometimes

Corresponding author
Alice C. McHardy,
alice.mchardy@helmholtz-hzi.de

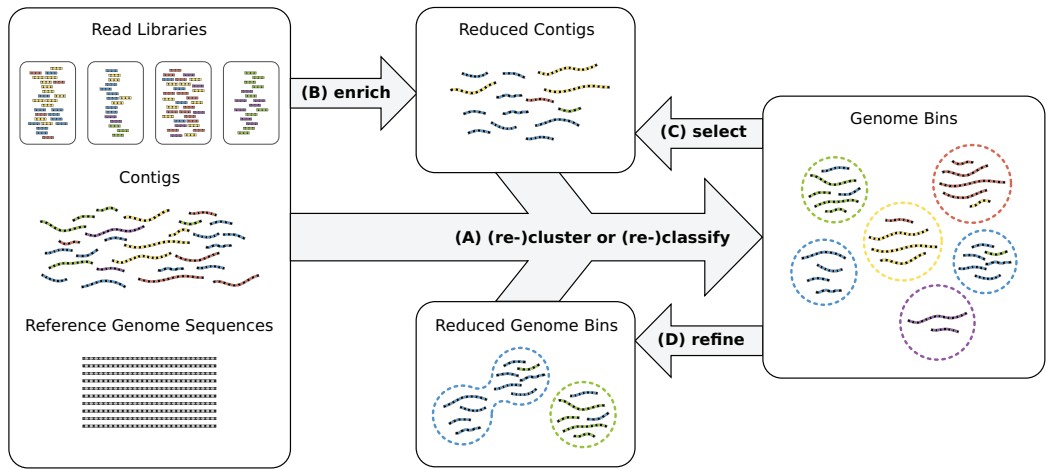

**Figure 1 Genome reconstruction workflow.** To recover genomes from environmental sequencing data, the illustrated processes can be iterated. Different programs can be run for each process and iteration. MGLEX can be applied in all steps: (A) to classify contigs or to cluster by embedding the probabilistic model into an iterative procedure; (B) to enrich a metagenome for a target genome to reduce its size and to filter out irrelevant sequence data; (C) to select contigs of existing bins based on likelihoods and $p$-values and to repeat the binning process with a reduced dataset; (D) to refine existing bins, for instance to merge bins as suggested by bin analysis.

using different sequencing protocols and technologies, such as Illumina and PacBio sequencing (*Hagen et al., 2016*). Genome copies are reflected by corresponding read coverage variation in the assemblies which allows to resolve samples with many genomes. The combination of experimental techniques helps to overcome platform-specific shortcomings such as short reads or high error rates in the data analysis. However, reconstructing high-quality bins of individual strains remains difficult without very high numbers of replicates. Often, genome reconstruction may improve by manual intervention and iterative analysis (Fig. 1) or additional sequencing experiments.

Genome bins can be constructed by consideration of genome-wide sequence properties. Currently, oftentimes the following types of information are considered:

- Read contig coverage: sequencing read coverage of assembled contigs, which reflects the genome copy number (organismal abundance) in the community. Abundances can vary across biological or technical replicates, and co-vary for contigs from the same genome, supplying more information to resolve individual genomes (*Baran & Halperin, 2012*; *Albertsen et al., 2013*).
- Nucleotide sequence composition: the frequencies of short nucleotide subsequences of length $k$ called $k$-mers. The genomes of different species have a characteristic $k$-mer spectrum (*Karlin, Mrazek & Campbell, 1997*; *McHardy et al., 2007*).
- Sequence similarity to reference sequences: a proxy for the phylogenetic relationship to species which have already been sequenced. The similarity is usually inferred by alignment to a reference collection and can be expressed using taxonomy (*McHardy et al., 2007*).

Probabilities represent a convenient and efficient way to represent and combine information that is uncertain by nature. Here, we:

- Propose a probabilistic aggregate model for binning based on three commonly used information sources, which can easily be extended to include new features.
- Outline the features and submodels for each information type. As the feature types listed above derive from distinct processes, we define for each of them independently a suitable probabilistic submodel.
- Showcase several applications related to the binning problem.

A model with data-specific structure poses an advantage for genome recovery in metagenomes because it uses data more efficiently for fragmented assemblies with short contigs or a low number of samples for differential coverage binning. Being probabilistic, it generates probabilities instead of hard labels so that a contig can be assigned to several, related genome bins and the uncertainty can easily be assessed. The models can be applied in different ways, not just classification, which we show in our application examples. Most importantly, there is a rich repertoire of higher-level procedures based on probabilistic models, including expectation maximization (EM) and Markov chain Monte Carlo (MCMC) methods for clustering without or with few prior knowledge of the modeled genomes.

We focus on defining explicit probabilistic models for each feature type and their combination into an aggregate model. In contrast, binning methods often concatenate and transform features (*Chatterji et al., 2008*; *Imelfort et al., 2014*; *Alneberg et al., 2014*) before clustering. Specific models for the individual data types can be better tailored to the data generation process and will therefore generally enable a better use of information and a more robust fit of the aggregate model while requiring fewer data. We propose a flexible model with regard to both the included features and the feature extraction methods. There already exist parametric likelihood models in the context of clustering, for a limited set of features. For instance, *Kislyuk et al. (2009)* use a model for nucleotide composition and *Wu et al. (2014)* integrated distance-based probabilities for 4-mers and absolute contig coverage using a Poisson model. We extend and generalize this work so that the model can be used in different contexts such as classification, clustering, genome enrichment and binning analysis. Importantly, we are not providing an automatic solution to binning but present a flexible framework to target problems associated with binning. This functionality can be used in custom workflows or programs for the steps illustrated in Fig. 1. As input, the model incorporates genome abundance, nucleotide composition and additionally sequence similarity (via taxonomic annotation). The latter is common as taxonomic binning output (*Dröge, Gregor & McHardy, 2014*; *Wood & Salzberg, 2014*; *Gregor et al., 2016*) and for quality assessment but has rarely been systematically used as features in binning (*Chatterji et al., 2008*; *Lu et al., 2017*). We show that taxonomic annotation is valuable information that can improve binning considerably.

## METHODS

### Classification models

Classification is a common concept in machine learning. Usually, such algorithms use training data for different classes to construct a model which then contains the condensed information about the important properties that distinguish the data of the classes. In probabilistic modeling, we describe these properties as parameters of likelihood functions, often written as $\boldsymbol{\theta}$. After $\boldsymbol{\theta}$ has been determined by training, the model can be applied to assign novel data to the modeled classes. In our application, classes are genomes, or bins, and the data are nucleotide sequences like contigs. Thus, contigs can be assigned to genomes bins but we need to provide training sequences for the genomes. Such data can be selected by different means, depending on the experimental and algorithmic context. One can screen metagenomes for genes which are unique to clades, or which can be annotated by phylogenetic approaches, and use the corresponding sequence data for training (*Gregor et al., 2016*). Independent assemblies or reference genomes can also serve as training data for genome bins (*Brady & Salzberg, 2009*; *Patil et al., 2011*; *Gregor et al., 2016*). Another direct application is to learn from existing genome bins, which were derived by any means, and then to (re)assign contigs to these bins. This is useful for short contigs which are often excluded from binning and analysis due to their high variability. Finally, probabilistic models can be embedded into iterative clustering algorithms with random initialization.

### Aggregate model

Let $1 \leq i \leq D$ be an index referring to $D$ contigs resulting from a shotgun metagenomic experiment. In the following we will present a generative probabilistic aggregate model that consists of components, indexed by $1 \leq k \leq M$, which are generative probabilistic models in their own right, yielding probabilities $P_k(\text{contig}_i)$ that $\text{contig}_i$ belongs to a particular genome. Each of the components $k$ reflects a particular feature such as:

- A weight $w_i$ (contig length)
- Sample abundance feature vectors $\boldsymbol{a_i}$ and $\boldsymbol{r_i}$, one entry per sample
- A compositional feature vector $\boldsymbol{c_i}$, one entry per compositional feature (e.g., a $k$-mer)
- A taxonomic feature vector $\boldsymbol{t_i}$, one entry per taxon

We define the individual feature vectors in the corresponding sections. As mentioned before, each of the $M$ features gives rise to a probability $P_k(\text{contig}_i \mid \text{genome})$ that $\text{contig}_i$ belongs to a specific genome by means of its component model. Those probabilities are then collected into an aggregate model that transforms those feature specific probabilities $P_k(i \mid \text{genome})$ into an overall probability $P(i \mid \text{genome})$ that contig $i$ is associated with the genome. In the following, we describe how we construct this model with respect to the individual submodels $P_k(i \mid \text{genome})$, the feature representation of the contigs and how we determine the optimal set of parameters from training sequences.

For the $i$th contig, we define a joint likelihood for genome bin $g$ (Eq. (1), the probabilities written as a function of the genome parameters), which is a weighted product over $M$ independent component likelihood functions, or submodels, for the different feature types. For the $k$th submodel, $\boldsymbol{\Theta}_k$ is the corresponding parameter vector, $\boldsymbol{F}_{ik}$ the feature vector of the $i$th contig and $\alpha_k$ defines the contribution of the respective submodel or feature type. $\beta$ is a free scaling parameter to adjust the smoothness of the aggregate likelihood distribution over the genome bins (bin posterior).

$$\mathcal{L}(\boldsymbol{\Theta_g} \mid \mathbf{F_i}) = \left( \prod_{k=1}^{M} \mathcal{L}(\boldsymbol{\Theta}_{gk} \mid \boldsymbol{F}_{ik})^{\alpha_k} \right)^{\beta} \tag{1}$$

We assume statistical independence of the feature subtypes and multiply likelihood values from the corresponding submodels. This is a simplified but reasonable assumption: e.g., the species abundance in a community can be altered by external factors without impacting the nucleotide composition of the genome or its taxonomic position. Also, there is no direct relation between a genome's $k$-mer distribution and taxonomic annotation via reference sequences.

All model parameters, $\boldsymbol{\Theta_g}$, $\boldsymbol{\alpha}$ and $\beta$, are learned from training sequences. We will explain later, how the weight parameters $\boldsymbol{\alpha}$ and $\beta$ are chosen and begin with a description of the four component likelihood functions, one for each feature type.

In the following, we denote the $j$th position in a vector $\boldsymbol{x_i}$ with $x_{i,j}$. To simplify notation, we also define the sum or fraction of two vectors of the same dimension as the positional sum or fraction and write the length of vector $\boldsymbol{x}$ as $len(\boldsymbol{x})$.

## Absolute abundance

We derive the average number of reads covering each contig position from assembler output or by mapping the reads back onto contigs. This mean coverage is a proxy for the genome abundance in the sample because it is roughly proportional to the genome copy number. A careful library preparation causes the copy numbers of genomes to vary differently over samples, so that each genome has a distinct relative read distribution. Depending on the amount of reads in each sample being associated with every genome, we obtain for every contig a coverage vector $\boldsymbol{a_i}$ where $len(\boldsymbol{a_i})$ is the number of samples. Therefore, if more sample replicates are provided, contigs from different genomes are generally better separable since every additional replicate adds an entry to the feature vectors.

Random sequencing followed by perfect read assembly theoretically produces positional read counts which are Poisson distributed, as described in *Lander & Waterman (1988)*. In Eq. (2), we derived a similar likelihood using mean coverage values (see Supplemental Information for details). The likelihood function is a normalized product over the independent Poisson functions $P_{\theta j}(a_{i,j})$ for each sample. The expectation parameter $\theta_j$ represents the genome copy number in the $j$th sample.

$$\mathcal{L}(\boldsymbol{\theta} \mid \boldsymbol{a_i}) = \sqrt[len(\boldsymbol{a_i})]{\prod_{j=1}^{len(\boldsymbol{a_i})} P_{\theta_j}(a_{i,j})} = \sqrt[len(\boldsymbol{a_i})]{\prod_{j=1}^{len(\boldsymbol{a_i})} \frac{\theta_j^{a_{i,j}}}{a_{i,j}!}e^{-\theta_j}} \tag{2}$$

The Poisson explicitly accounts for low and zero counts, unlike a Gaussian model. Low counts are often observed for undersequenced and rare taxa. Note that $a_{i,j}$ is independent of $\boldsymbol{\theta}$. We derived the model likelihood function from the joint Poisson over all contig positions by approximating the first data-term with mean coverage values (Supplemental Information).

The maximum likelihood estimate (MLE) for $\boldsymbol{\theta}$ on training data is the weighted average of mean coverage values for each sample in the training data (Supplemental Information).

$$\hat{\boldsymbol{\theta}} = \frac{\sum_{i=1}^{N} w_i \boldsymbol{a_i}}{\sum_{i=1}^{N} w_i} \tag{3}$$

### Relative abundance

In particular for shorter contigs, the absolute read coverage is often overestimated. Basically, the Lander–Waterman assumptions (*Lander & Waterman, 1988*) are violated if reads do not map to their original locations due to sequencing errors or if they "stack" on certain genome regions because they are ambiguous (i.e., for repeats or conserved genes), rendering the Poisson model less appropriate. The Poisson, when constrained on the total sum of coverages in all samples, leads to a binomial distribution as shown by *Przyborowski & Wilenski (1940)*. Therefore, we model differential abundance over different samples using a binomial in which the parameters represent a relative distribution of genome reads over the samples. For instance, if a particular genome had the same copy number in a total of two samples, the genome's parameter vector $\boldsymbol{\theta}$ would simply be [0.5,0.5]. As for absolute abundance, the model becomes more powerful with a higher number of samples. Using relative frequencies as model parameters instead of absolute coverages, however, has the advantage that any constant coverage factor cancels in the division term. For example, if a genome has two similar gene copies which are collapsed during assembly, twice as many reads will map onto the assembled gene in every sample but the relative read frequencies over samples will stay unaffected. This makes the binomial less sensitive to read mapping artifacts but requires two or more samples because one degree of freedom (DF) is lost by the division.

The contig features $\boldsymbol{r_i}$ are the mean coverages in each sample, which is identical to $\boldsymbol{a_i}$ in the absolute abundance model, and the model's parameter vector $\boldsymbol{\theta}$ holds the relative read frequencies in the samples, as explained before. In Eq. (4), we ask: how likely is the observed mean contig coverage $r_{i,j}$ in sample $j$ given the genome's relative read frequency $\theta_j$ of the sample and the contig's total coverage $R_i$ for all samples. The corresponding

likelihood is calculated as a normalized product over the binomials $B_{R_i,\theta_j}(r_{i,j})$ for every sample.

$$\mathcal{L}(\boldsymbol{\theta} \mid \boldsymbol{r_i}) = \sqrt[len(\boldsymbol{r_i})]{\prod_{j=1}^{len(\boldsymbol{r_i})} B_{\boldsymbol{R_i},\theta_j}(r_{i,j})} = \sqrt[len(\boldsymbol{r_i})]{\prod_{j=1}^{len(\boldsymbol{r_i})} \binom{R_i}{r_{i,j}} \theta_j^{r_{i,j}} (1-\theta_j)^{(R_i - r_{i,j})}} \qquad (4)$$

$R_i$ is the sum of the abundance vector $\boldsymbol{r_i}$. Because both $R_i$ and $r_i$ can contain real numbers, we need to generalize the binomial coefficient to positive real numbers via the gamma function $\Gamma$.

$$\binom{n}{k} = \frac{\Gamma(n+1)\Gamma(k+1)}{\Gamma(n-k+1)} \qquad (5)$$

Because the binomial coefficient is a constant factor and independent of $\boldsymbol{\theta}$, it can be omitted in ML classification (when comparing between different genomes) or be retained upon parameter updates. As for the Poisson, the model accounts for low and zero counts (by the binomial coefficient). We derived the likelihood function from the joint distribution over all contig positions by approximating the binomial data-term with mean coverage values (see Supplemental Information).

The MLE $\hat{\boldsymbol{\theta}}$ for the model parameters on training sequence data corresponds to the amount of read data (base pairs) in each sample divided by the total number of base pairs in all samples. We express this as a weighted sum of contig mean coverage values (see Supplemental Information).

$$\hat{\boldsymbol{\theta}} = \frac{\sum_{i=1}^{N} w_i \boldsymbol{r_i}}{\sum_{i=1}^{N} w_i R_i} \qquad (6)$$

It is obvious that absolute and relative abundance models are not independent when the identical input vectors (here $\boldsymbol{a_i} = \boldsymbol{r_i}$) are used. However, we can instead apply the Poisson model to the total coverage $R_i$ (summed over all samples) because this sum also follows a Poisson distribution. To illustrate the total abundance, this compares to mixing the samples before sequencing so that the resolution of individual samples is lost. The binomial, in contrast, only captures the relative distribution of reads over the samples (one DF is lost in the ratio transform). This way, we can combine both absolute and relative abundance submodels in the aggregate model.

## Nucleotide composition

Microbial genomes have a distinct "genomic fingerprint" (*Karlin, Mrazek & Campbell, 1997*) which is typically determined by means of $k$-mers. Each contig has a relative frequency vector $\boldsymbol{c_i}$ for all possible $k$-mers of size $k$. The nature of shotgun sequencing

demands that each $k$-mer is counted equally to its reverse complement because the orientation of the sequenced strand is typically unknown. With increasing $k$, the feature space grows exponentially and becomes sparse. Thus, it is common to select $k$ from 4 to 6 (*Teeling et al., 2004*; *McHardy et al., 2007*; *Kislyuk et al., 2009*). Here, we simply use 5-mers ($len(c_i) = \frac{4^5}{2} = 512$) but other choices can be made.

For its simplicity and effectiveness, we chose a likelihood model assuming statistical independence of features so that the likelihood function in Eq. (7) becomes a simple product over observation probabilities (or a linear model when transforming into a log-likelihood). Though $k$-mers are not independent due to their overlaps and reverse complementarity (*Kislyuk et al., 2009*), the model has been successfully applied to $k$-mers (*Wang et al., 2007*), and we can replace $k$-mers in our model with better-suited compositional features. A genome's background distribution $\boldsymbol{\theta}$ is a vector which holds the probabilities to observe each $k$-mer and the vector $\boldsymbol{c_i}$ does the same for the $i$th contig. The composition likelihood for a contig is a weighted and normalized product over the background frequencies.

$$\mathcal{L}(\boldsymbol{\theta} \mid \boldsymbol{c_i}) = \prod_{i=1}^{len(c_i)} \theta_i^{c_i} \tag{7}$$

The genome parameter vector $\hat{\boldsymbol{\theta}}$ that maximizes the likelihood on training sequence data can be estimated by a weighted average of feature counts (Supplemental Information).

$$\hat{\boldsymbol{\theta}} = \frac{\sum_{i=1}^{N} w_i \boldsymbol{c_i}}{\sum_{i=1}^{N} w_i} \tag{8}$$

## Similarity to reference

We can compare contigs to reference sequences, for instance by local alignment. Two contigs that align to closely related taxa are more likely to derive from the same genome than sequences which align to distant clades. We convert this indirect relationship to explicit taxonomic features which we can compare without direct consideration of reference sequences. A taxon is a hierarchy of nested classes which can be written as a tree path, for example, the species *Escherichia coli* could be written as "Bacteria, Gammaproteobacteria, Enterobacteriaceae, and *E. coli*".

We assume that distinct regions of a contig, such as genes, can be annotated with different taxa. Each taxon has a corresponding weight which in our examples is a positive alignment score. The weighted taxa define a spectrum over the taxonomy for every contig and genome. It is not necessary that the alignment reference be complete or include the respective species genome but all spectra must be equally biased. Since each contig is represented by a hierarchy of $L$ numeric weights, we incorporated these features into our multi-layer model. First, each contig's taxon weights are transformed to a set of sparse

**Table 1 Calculating the contig features $t_i$ for a simplified taxonomy.**

| Node | Taxon | Level $l$ | Index $j$ | Score | $t_{i,l,j}$ |
|------|-------|-----------|-----------|-------|-------------|
| a | Bacteria | 1 | 1 | 0 | 7 |
| b | Gammaproteobacteria | 2 | 1 | 0 | 6 |
| c | Betaproteobacteria | 2 | 2 | 1 | 1 |
| d | Enterobacteriaceae | 3 | 1 | 0 | 5 |
| e | Yersiniaceae | 3 | 2 | 1 | 1 |
| f | *E. vulneris* | 4 | 1 | 1 | 1 |
| g | *E. coli* | 4 | 2 | 3 | 3 |
| h | *Yersinia sp.* | 4 | 3 | 1 | 1 |

**Note:**
There are five original integer alignment scores for nodes (c), (e), (f), (g), and (h) which are summed up at higher levels to calculate the feature vectors $t_{i,l}$. The corresponding tree structure is shown in Fig. 2.

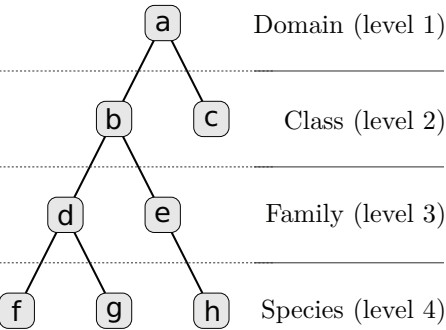

**Figure 2 Taxonomy for Table 1 which is simplified to four levels and eight nodes.** A full taxonomy may consist of thousands of nodes.

feature vectors $t_i = \{t_{i,l} \mid 1 \leq l \leq L\}$, one for each taxonomic level, by inheriting and accumulating scores for higher-level taxa (see Table 1; Fig. 2).

Each vector $t_{i,l}$ contains the scores for all $T_l$ possible taxa at level $l$. A genome is represented by a similar set of vectors $\theta = \{\theta_l \mid 1 \leq l \leq L\}$ with identical dimensions, but here, entries represent relative frequencies on the particular level $l$, for instance a distribution over all family taxa. The corresponding likelihood model corresponds to a set of simple frequency models, one for each layer. The full likelihood is a product of the level likelihoods.

$$\mathcal{L}(\theta \mid t_i) = \prod_{l=1}^{L} \prod_{j=1}^{T_l} \theta_{l,j}^{t_{i,l,j}} \tag{9}$$

For simplicity, we assume that layer likelihoods are independent which is not quite true but effective. The MLE for each $\theta_l$ is then derived from training sequences similar to the simple frequency model (Supplemental Information).

$$\hat{\theta}_l = \frac{\sum_{i=1}^{N} t_{i,l}}{\sum_{j=1}^{T_l} \sum_{i=1}^{N} t_{i,l}} \tag{10}$$

## Inference of weight parameters

The aggregate likelihood for a contig in Eq. (1) is a weighted product of submodel likelihoods. The weights in vector $\boldsymbol{\alpha}$ balance the contributions, assuming that they must not be equal. When we write the likelihood in logarithmic form (Eq. (11)), we see that each weight $\alpha_k$ sets the variance or width of the contigs' submodel log-likelihood distribution. We want to estimate $\alpha_k$ in a way which is not affected by the original submodel variance because the corresponding normalization exponent is somewhat arbitrary. For example, we normalized the nucleotide composition likelihood as a single feature and the abundance likelihoods as a single sample to limit the range of the likelihood values, because we simply cannot say how much each feature type counts.

$$l(\boldsymbol{\Theta} \mid \mathbf{F_i}) = \beta \sum_{k=1}^{M} \alpha_k l(\boldsymbol{\Theta}_k \mid \boldsymbol{F}_{i,k}) \qquad (11)$$

For any modeled genome, each of the $M$ submodels produces a distinct log-likelihood distribution of contig data. Based on the origin of the contigs, which is known for model training, the distribution can be split into two parts, the actual genome (positive class) and all other genomes (negative class), as illustrated in Fig. 3. The positive distribution is roughly unimodal and close to zero whereas the negative distribution, which represents many genomes at once, is diverse and yields strongly negative values. Intuitively, we want to select $\boldsymbol{\alpha}$ such that the positive class is well separated from the negative class in the aggregate log-likelihood function in Eq. (11).

Because $\boldsymbol{\alpha}$ cannot be determined by likelihood maximization, the contributions are balanced in a robust way by setting $\boldsymbol{\alpha}$ to the inverse standard deviation of the genome (positive class) log-likelihood distributions. More precisely, we calculate the average standard deviation over all genomes weighted by the amount of contig data (bp) for each genome and calculate $\alpha_k$ as the inverse of this value. This scales down submodels with a high average variance. When we normalize the standard deviation of genome log-likelihood distributions in all submodels before summation, we assume that a high variance means uncertainty. This form of weight estimation requires that for at least some of the genomes, a sufficient number of sequences must be available to estimate the standard deviation. In some instances, it might be necessary to split long contigs into smaller sequences to generate a sufficient number of data points for estimation.

Parameter $\beta$ in Eq. (11) is only relevant for soft classification but not in the context of ML classification or $p$-values. It can best be viewed as a sharpening or smoothing parameter of the bin posterior distribution (the probability of a genome or bin given the contig). $\beta$ is estimated by minimization of the training or test error, as in our simulation.

## Data simulation

We simulated reads of a complex microbial community from 400 publicly available genomes (Supplemental Information and Table S1). These comprised 295 unique and 44 species with each two or three strain genomes to mimic strain heterogeneity.

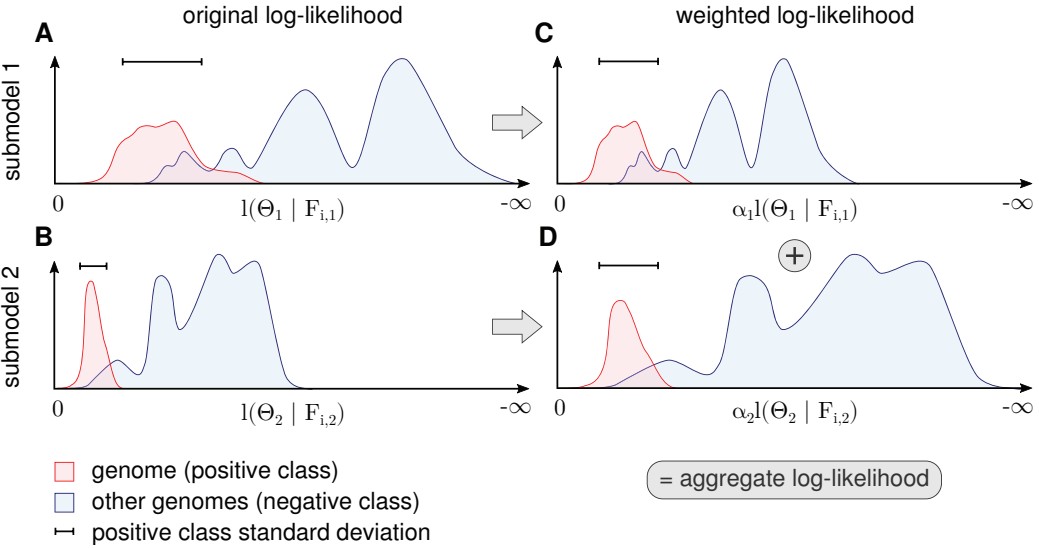

**Figure 3** **Procedure for determination of $\alpha_k$ for each submodel.** The figure shows a schematic for a single genome and two submodels. The genome's contig log-likelihood distribution (A and B) is scaled to a standard deviation of one (C and D) before adding the term in the aggregate model in Eq. (11).

Our aim was to create a difficult benchmark dataset under controlled settings, minimizing potential biases introduced by specific software. We sampled abundances from a lognormal distribution because it has been described as a realistic model (*Schloss & Handelsman, 2006*). We then simulated a primary community which was then subject to environmental changes resulting in exponential growth of 25% of the community members at growth rates which where chosen uniformly at random between one and ten whereas the other genome abundances remained unchanged. We applied this procedure three times to the primary community which resulted in one primary and three secondary artificial community abundances profiles. With these, we generated 150 bp long Illumina HiSeq reads using the ART simulator (*Huang et al., 2012*) and chose a yield of 15 Gb per sample. The exact amount of read data for all four samples after simulation was 59.47 Gb. To avoid any bias caused by specific metagenome assembly software and to assure a constant contig length, we divided the original genome sequences into non-overlapping artificial contigs of 1 kb length and selected a random 500 kb of each genome to which we mapped the simulated reads using Bowtie2 (*Langmead & Salzberg, 2012*). By the exclusion of some genome reference, we imitated incomplete genome assemblies when mapping reads, which affects the coverage values. Finally, we subsampled 300 kb contigs per genome with non-zero read coverage in at least one of the samples to form the demonstration dataset (120 Mb), which has 400 genomes (including related strains), four samples and contigs of size 1 kb. Due to the short contigs and few samples, this is a challenging dataset for complete genome recovery (*Nielsen et al., 2014*) but suitable to demonstrate the functioning of our model with limited data. For each contig we derived 5-mer frequencies, taxonomic annotation (removing species-level genomes from the reference sequence data) and average read coverage per sample, as described in Supplemental Information.

## RESULTS

### Maximum likelihood classification

We evaluated the performance of the model when classifying contigs to the genome with the highest likelihood, a procedure called maximum likelihood (ML) classification. We applied a form of three-fold cross-validation, dividing the simulated data set into three equally-sized parts with 100 kb from every genome. We used only 100 kb (training data) of every genome to infer the model parameters and the other 200 kb (test data) to measure the classification error. A total of 100 kb was used for training because it is often difficult to identify sufficient training data in metagenome analysis. For each combination of submodels, we calculated the mean squared error (MSE) and mean pairwise coclustering (MPC) probability for the predicted (ML) probability matrices (Supplemental Information), averaged over the three test data partitions. We included the MPC as it can easily be interpreted: for instance, a value of 0.5 indicates that on average 50% of all contig pairs of a genome end up in the same bin after classification. Table 2 shows that the model integrates information from each data source such that the inclusion of additional submodels resulted in a better MPC and also MSE, with a single exception when combining absolute and relative abdundance models which resulted in a marginal increase of the MSE. We also found that taxonomic annotation represents the most powerful information type in our simulation. For comparison, we added scores for NBC (*Rosen, Reichenberger & Rosenfeld, 2011*), a classifier based on nucleotide composition with in-sample training using 5-mers and 15-mers, and centrifuge (*Kim et al., 2016*), a similarity-based classifier both with in-sample and reference data. These programs were given the same information as the corresponding submodels and they rank close to these. In a further step, we investigated how the presence of very similar genomes impacted the performance of the model. We first collapsed strains from the same species by merging the corresponding columns in the classification likelihood matrix, retaining the entry with the highest likelihood, and then computed the resulting coclustering performance increase $\Delta\text{MPC}_{\text{ML}}$. Considering assignment on species instead of strain level showed a larger $\Delta\text{MPC}_{\text{ML}}$ for nucleotide composition and taxonomic annotation than for absolute and relative abundance. This is expected, because both do not distinguish among strains, whereas genome abundance does in some, but not all cases.

### Soft assignment

The contig length of 1 kb in our simulation is considerably shorter, and therefore harder to classify, than sequences which can be produced by current assembly methods or by some cutting-edge sequencing platforms (*Goodwin, McPherson & McCombie, 2016*). In practice, longer contigs can be classified with higher accuracy than short ones, as more information is provided as a basis for assignment. For instance, a more robust coverage mean, a $k$-mer spectrum derived from more counts or more local alignments to reference genomes can be inferred from longer sequences. However, as short contigs remain frequent in current metagenome assemblies, 1 kb is sometimes considered a minimum useful contig length (*Alneberg et al., 2014*). To account for the natural

**Table 2 Cross-validation performance of ML classification for all possible combinations of submodels.**

| Submodels | $MPC_{ML}$ | $\Delta MPC_{ML}$ | $MSE_{ML}$ |
|---|---|---|---|
| *Centrifuge (in-sample)* | *0.01* | 0.01 | 0.51 |
| *NBC (15-mers)* | 0.02 | *+0.00* | *0.66* |
| AbAb | 0.03 | +0.00 | 0.58 |
| ReAb | 0.08 | +0.02 | 0.61 |
| *Centrifuge (reference)* | 0.13 | +0.03 | 0.45 |
| AbAb + ReAb | 0.21 | +0.04 | 0.59 |
| NuCo | 0.30 | +0.06 | 0.52 |
| *NBC (5-mers)* | 0.34 | +0.06 | 0.48 |
| ReAb + NuCo | 0.41 | +0.07 | 0.48 |
| AbAb + NuCo | 0.43 | +0.08 | 0.50 |
| TaAn | 0.46 | +0.09 | 0.41 |
| AbAb + ReAb + NuCo | 0.52 | +0.09 | 0.44 |
| NuCo + TaAn | 0.52 | +0.09 | 0.40 |
| AbAb + TaAn | 0.54 | +0.09 | 0.39 |
| AbAb + NuCo + TaAn | 0.60 | +0.10 | 0.37 |
| ReAb + TaAn | 0.60 | +0.10 | 0.36 |
| ReAb + NuCo + TaAn | 0.64 | **+0.11** | 0.34 |
| AbAb + ReAb + TaAn | 0.65 | +0.10 | 0.35 |
| AbAb + ReAb + NuCo + TaAn | **0.68** | **+0.11** | **0.33** |

Notes:

We calculated the mean pairwise coclustering (MPC), the strain to species MPC improvement ($\Delta MPC_{ML}$) and the mean squared error (MSE). NBC (v1.1) and Centrifuge (v.1.0.3b) are external classifiers added for comparison. Best values are in bold and worst in italic. AbAb, absolute total abundance; ReAb, relative abundance; NuCo, nucleotide composition; TaAn, taxonomic annotation.

uncertainty when assigning short contigs, one can calculate the posterior probabilities over the genomes (see Supplemental Information), which results in partial assignments of each contig to the genomes. This can reflect situations in which a particular contig is associated with multiple genomes, for instance in case of misassemblies or the presence of homologous regions across genomes.

The free model parameter $\beta$ in Eq. (1), which is identical in all genome models, smoothens or sharpens the posterior distribution: $\beta = 0$ produces a uniform posterior and with very high $\beta$, the posterior approaches the sharp ML solution. We determined $\beta$ by optimizing the MSE on both training and test data, shown in Fig. 4. As expected, the classification training error was smaller than the test error because the submodel parameters were optimized with respect to the training data. Because the minima are close to each other, the full aggregate model seems robust to overfitting of $\beta$ on training data. The comparison of soft vs. hard assignment shows that the former has a smaller average test classification MSE of ~0.28 (the illustrated minimum in Fig. 4) compared to the latter (ML) assignment MSE of ~0.33 in Table 2. Thus, soft assignment seems more suitable to classify 1 kb contigs, which tend to produce similar likelihoods under more than one genome model.

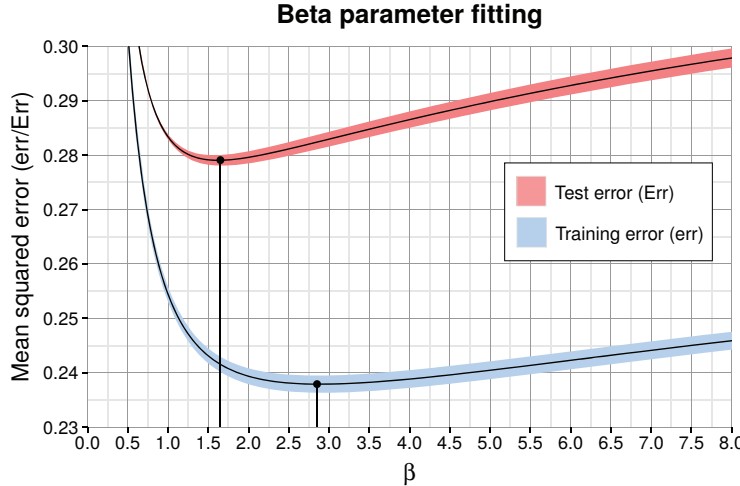

**Figure 4 Model training (err) and test error (Err) as a function of $\beta$ for the complete aggregate model including all submodels and feature types.** The solid curve shows the average and the colored shading the standard deviation of the three partitions in cross-validation. The corresponding optimal values for $\beta$ are marked by black dots and vertical lines. The minimum average training error is 0.238 ($\beta = 2.85$) and test error is 0.279 at $\beta = 1.65$.

## Genome enrichment

Enrichment is commonly known as an experimental technique to increase the concentration of a target substance relative to others in a probe. Thus, an enriched metagenome still contains a mixture of different genomes, but the target genome will be present at much higher frequency than before. This allows a more focused analysis of the contigs or an application of methods which seem prohibitive for the full data by runtime or memory considerations. In the following, we demonstrate how to filter metagenome contigs by $p$-value to enrich *in silico* for specific genomes. Often, classifiers model an exhaustive list of alternative genomes but in practice it is difficult to recognize all species or strains in a metagenome with appropriate training data. When we only look at individual likelihoods, for instance the maximum among the genomes, this can be misleading if the contig comes from a missing genome. For better judgment, a $p$-value tells us how frequent or extreme the actual likelihood is for each genome. Many if not all binning methods lack explicit significance calculations. We can take advantage of the fact that the classification model compresses all features into a genome likelihood and generate a null (log-)likelihood distribution on training data for each genome. Therefore, we can associate empirical $p$-values with each newly classified contig and can, for sufficiently small $p$-values, reject the null hypothesis that the contig belongs to the respective genome. Since this is a form of binary classification, there is the risk to reject a good contig which we measure as sensitivity.

We enriched a metagenome by first training a genome model and then calculating the $p$-values of remaining contigs using this model. Contigs with higher $p$-values than the chosen critical value were discarded. The higher this cutoff is, the smaller the enriched sample becomes, but also the target genome will be less complete. We calculated the reduced sample size as a function of the $p$-value cutoff for our simulation (Fig. 5).

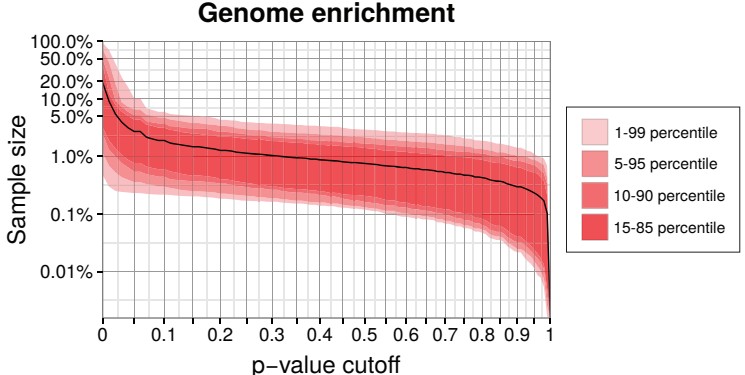

**Figure 5 Genome enrichment for 400 genomes with three-fold cross-validation.** For each genome, we measured the test sample size relative to the full dataset after filtering by a *p*-value cutoff and summing over the three data partitions. The solid line shows the resulting average sample size over all 400 genomes. The variability between genomes is shown as quantiles in red. Both axes are logarithmic to show the relevant details for lower *p*-values cutoffs. The corresponding sensitivity, shown in Fig. S1, is approximately a linear function of the *p*-value.

Selecting a *p*-value threshold of 2.5% shrinks the test data on average down to 5% of the original size. Instead of an empirical *p*-value, we could also use a parametrized distribution or select a critical log-likelihood value by manual inspection of the log-likelihood distribution (see Fig. 3 for an example of such a distribution). This example shows that generally a large part of a metagenome dataset can be discarded while retaining most of the target genome sequence data.

## Bin analysis

The model can be used to analyze bins of metagenome contigs, regardless of the method that was used to infer these bins. Specifically, one can measure the similarity of two bins in terms of the contig likelihood instead of, for instance, an average euclidean distance based on the contig or genome *k*-mer and abundance vectors. We compare bins to investigate the relation between the given data, represented by the features in the model, and their grouping into genome bins. For instance, one could ask whether the creation of two genome bins is sufficiently backed up by the contig data or whether they should be merged into a single bin. For readability, we write the likelihood of a contig in bin A to:

$$L(\theta_A \,|\, \text{contig } i) = L_i(\theta_A) = L(\theta_A) = L_A$$

To compare two specific bins, we select the corresponding pair of columns in the classification likelihood matrix and calculate two mixture likelihoods for each contig (rows), $\hat{L}$, using the MLE of the parameters for both bins and $L_{swap}$ under the hypothesis that we swap the model parameters of both bins. The partial assignment weights $\hat{\pi}_A$ and $\hat{\pi}_B$, called responsibilities, are estimated by normalization of the two bin likelihoods.

$$\hat{L} = \hat{\pi}_A L_A + \hat{\pi}_B L_B = \left(\frac{L_A}{L_A + L_B}\right) L_A + \left(\frac{L_B}{L_A + L_B}\right) L_B = \frac{L_A^2 + L_B^2}{L_A + L_B} \tag{12}$$

$$L_{swap} = \hat{\pi}_A L_B + \hat{\pi}_B L_A = \left(\frac{L_A}{L_A + L_B}\right) L_B + \left(\frac{L_B}{L_A + L_B}\right) L_A = \frac{2 L_A L_B}{L_A + L_B} \tag{13}$$

For example, if $\hat{\pi}_A$ and $\hat{\pi}_B$ assign one third of a contig to the first, less likely bin and two thirds to the second, more likely bin using the optimal parameters, then $L_{swap}$ would simply exchange the contributions in the mixture likelihood so that one third are assigned to the more likely and two thirds to the less likely bin. The ratio $L_{swap}/\hat{L}$ ranges from 0 to 1 and can be seen as a percentage similarity. We form a joint relative likelihood for all $N$ contigs, weighting each contig by its optimal mixture likelihood $\hat{L}$ and normalizing over these likelihood values.

$$S(A, B) = \sqrt[z]{\prod_{i=1}^{N} \left(\frac{2 L_i(\theta_A) L_i(\theta_B)}{L_i^2(\theta_A) + L_i^2(\theta_B)}\right)^{\frac{L_i^2(\theta_A) + L_i^2(\theta_B)}{L_i(\theta_A) + L_i(\theta_B)}}} \tag{14}$$

normalized by the total joint mixture likelihood

$$Z = \sum_{i=1}^{N} \frac{L_i^2(\theta_A) + L_i^2(\theta_B)}{L_i(\theta_A) + L_i(\theta_B)} \tag{15}$$

The quantity in Eq. (14) ranges from 0 to 1, reaching one when the two bin models produce identical likelihood values. We can therefore interpret the ratio as a percentage similarity between any two bins. A connection to the Kullback–Leibler divergence can be constructed (Supplemental Information).

To demonstrate the application, we trained the model on our simulated genomes, assuming they were bins, and created trees (Fig. 6) for a randomly drawn subset of 50 of the 400 genomes using the probabilistic bin distances $-log(S)$ (Eq. (14)). We computed the distances twice, first with only nucleotide composition and taxonomic annotation submodels and second with the full feature set to compare the bin resolution. The submodel parameters were inferred using the full dataset and $\beta$ using three-fold crossvalidation. We then applied average linkage clustering to build balanced and rooted trees with equal distance from leave to root for visual inspection. The first tree loosely reflects phylogenetic structure corresponding to the input features. However, many similarities over 50% (outermost ring) show that model and data lack the support for separating these bins. In contrast, the fully informed tree, which additionally includes information about contig coverages, separates the genomes bins, such that only closely related strains remain ambiguous. This analysis shows again that the use of additional features improves the resolution of individual genomes and, specifically, that abundance separates similar genomes. Most importantly, we show that our model provides a measure of support for a genome binning. We know the taxa of the genome bins in

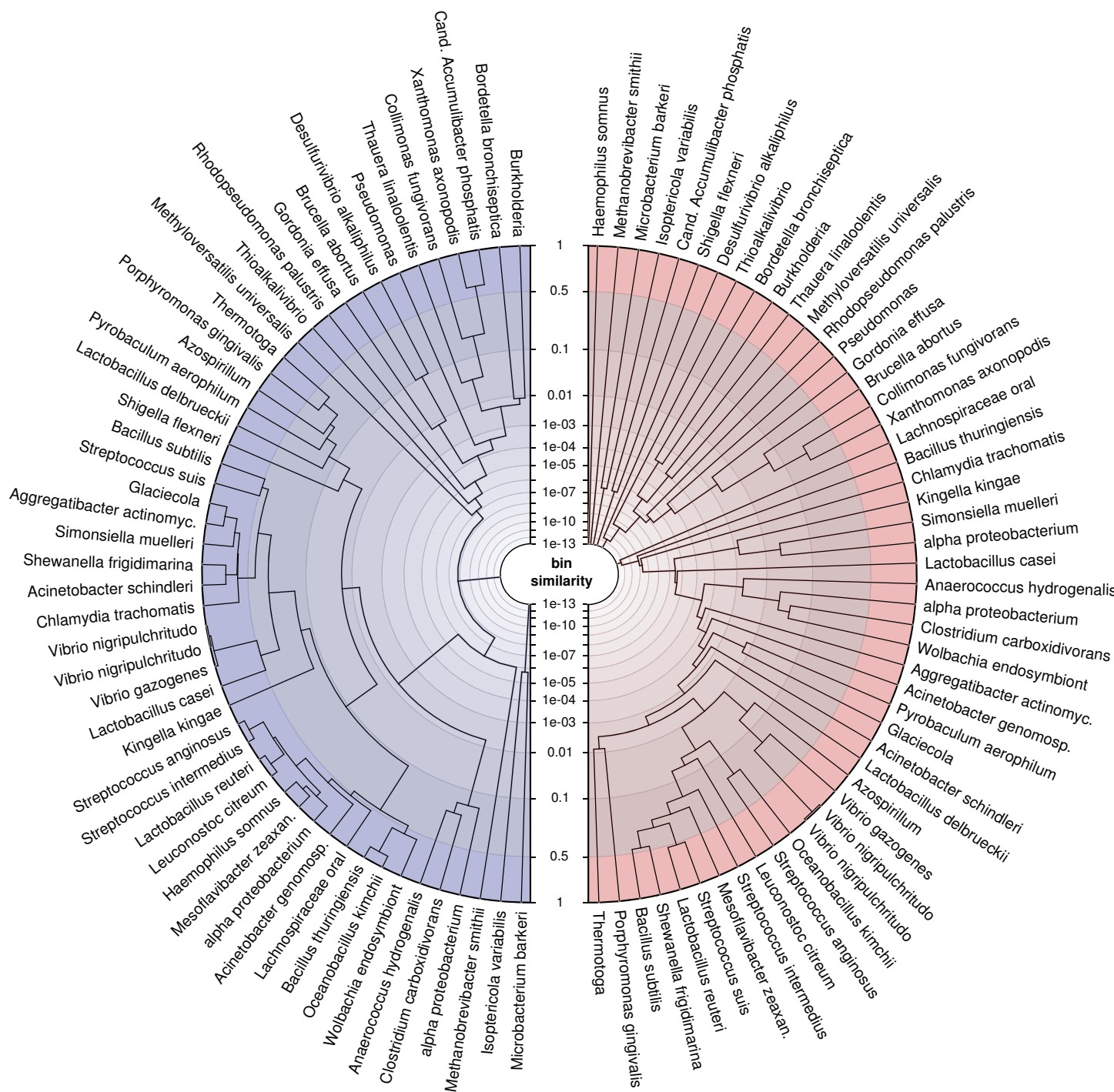

**Figure 6 Average linkage clustering of a random subset of 50 out of 400 genomes using probabilistic distances** $-log(S)$ **(Eq. (14)) to demonstrate the ability of the model to measure bin resolution.** This example compares the left (blue) tree, which was constructed only with nucleotide composition and taxonomic annotations, with the right (red) tree, which uses all available features. The tip labels were shortened to fit into the figure. The similarity axis is scaled as $log(1-log(S))$ to focus on values near one. Bins which are more than 50% similar branch in the outermost ring whereas highly dissimilar bins branch close to the center. We created the trees by applying the R function *hclust*(method = "average") to MGLEX output.

**Table 3 Genome bin refinement for CAMI medium complexity dataset with 232 genomes and two samples.**

| Binner | Variant | Bin count | Recall (bp) | ARI |
|---|---|---|---|---|
| Metawatt | Unmodified | 285 | *0.94* | *0.75* |
| Metawatt | MGLEX swapped contigs | 285 | *0.94* | **0.82** |
| Metawatt | MGLEX all contigs | 285 | **1.00** | 0.77 |
| MaxBin | Unmodified | 125 | *0.82* | 0.90 |
| MaxBin | MGLEX swapped contigs | 125 | *0.82* | **0.92** |
| MaxBin | MGLEX all contigs | 125 | **1.00** | *0.76* |

**Note:**
The recall is the fraction of overall assigned contigs (bp). The adjusted rand index (ARI) is a measure of binning precision. The unmodified genome bins are the submissions to the CAMI challenge using the corresponding unsupervised binning methods Metawatt and MaxBin. MGLEX swapped contigs: contigs in original genome bins reassigned to the bin with highest MGLEX likelihood. MGLEX all contigs: all contigs (with originally uncontained) assigned to the bin with highest MGLEX likelihood. The lowest scores are written in italic and highest in bold.

this example but for real metagenomes, such an analysis can reveal binning problems and help to refine the bins as in Fig. 1D.

## Genome bin refinement

We applied the model to show one of its current use cases on more realistic data. We downloaded the medium complexity dataset from https://www.cami-challenge.org. This dataset is quite complex (232 genomes, two sample replicates). We also retrieved the results of two highest-performing automatic binning programs, MaxBin and Metawatt, in the CAMI challenge evaluation (*Sczyrba et al., 2017*). We took the simplest possible approach: we trained MLGEX on the genome bins derived by these methods and classified the contigs to the bins with the highest likelihood, thus ignoring all details of contig splitting, $\beta$ or $p$-value calculation and the possibility of changing the number of genome bins. When contigs were assigned to multiple bins with equal probability, we attributed them to the first bin in the list because the CAMI evaluation framework did not allow sharing contigs between bins. In our evaluation, we only used information provided to the contestants by the time of the challenge. We report the results for two settings for each method using the recall, the fraction of overall assigned contigs (bp), and the adjusted rand index (ARI) as defined in *Sczyrba et al. (2017)*. Both measures are dependent so that usually a tradeoff between them is chosen. In the first experiment, we swapped contigs which were originially assigned between bins. In the second experiment, all available contigs were assigned, thus maximizing the recall. Table 3 shows that MGLEX bin refinement improved the genome bins in terms of the ARI for both sets of genome bins when fixing the recall, and increased in both measures for Metawatt but not MaxBin when assigning all contigs including the originally unassigned. This is likely due to the fact that MaxBin has fewer but relatively complete bins to which the other contigs cannot correctly be recruited. Further improvement would involve dissecting and merging bins within and among methods, for which MGLEX likelihoods can be considered.

## Implementation

We provide a Python package called MGLEX, which includes the described model. Simple text input facilitates the integration of external programs for feature extraction like *k*-mer counting or read mapping, which are not included. MGLEX can process millions of sequences with vectorized arithmetics using NumPy (*Van der Walt, Colbert & Varoquaux, 2011*) and includes a command line interface to the main functionality, such as model training, classification, *p*-value and error calculations. It is open source (GPLv3) and freely available via the *Python Package Index* (https://pypi.python.org/pypi/mglex/) and on *GitHub* (https://www.github.com/hzi-bifo/mglex/).

# DISCUSSION

We describe an aggregate likelihood model for the reconstruction of genome bins from metagenome data sets and show its value for several applications. The model can learn from and classify nucleotide sequences from metagenomes. It provides likelihoods and posterior bin probabilities for existing genome bins, as well as *p*-values, which can be used to enrich a metagenome dataset with a target genome. The model can also be used to quantify bin similarity. It builds on four different submodels that make use of different information sources in metagenomics; namely, absolute and relative contig coverage, nucleotide composition and previous taxonomic assignments. By its modular design, the model can easily be extended to include additional information sources. This modularity also helps in interpretation and computations. The former, because different features can be analyzed separately and the latter, because submodels can be trained independently and in parallel.

In comparison to previously described parametric binning methods, our model incorporates two new types of features. The first is relative differential coverage, for which, to our knowledge, this is the first attempt to use binomials to account for systematic bias in the read mapping for different genome regions. As such, the binomial submodel represents the parametric equivalent of covariance distance clustering. The second new type is taxonomic annotation, which substantially improved the classification results in our simulation. Taxonomic annotations, as used in the model and in our simulation, were not correct up to the species level and need not be, as seen in the classification results. We only require the same annotation method be applied to all sequences. In comparison to previous methods, our aggregate model has weight parameters to combine the different feature types and allows tuning the bin posterior distribution by selection of an optimal smoothing parameter *β*.

We showed that probabilistic models represent a good choice to handle metagenomes with short contigs or few sample replicates, because they make soft, not hard decisions, and because they can be applied in numerous ways. When the individual submodels are trained, genome bin properties are compressed into fewer model parameters, such as mean values, which are mostly robust to outliers and therefore tolerate a certain fraction of bin pollution. This property allows to reassign contigs to bins, which we demonstrated in the "Genome bin refinement" section. Measuring the performance of the individual submodels and their corresponding features on short simulated contigs

(Table 2), we find that they discriminate genomes or species pan-genomes by varying degrees. Genome abundance represents, in our simulation with four samples, the weakest single feature type, which will likely become more powerful with increasing sample numbers. Notably, genomes of individual strains are more difficult to distinguish than species level pangenomes using any of the features. In practice, if not using idealized assemblies as in our current evaluation, strain resolution poses a problem to metagenome assembly, which is currently not resolved in a satisfactory manner (*Sczyrba et al., 2017*).

The current MGLEX model is somewhat crude because it makes many simplifying assumptions in the submodel definitions. For instance, the multi-layer model for taxonomic annotation assumes that the probabilities in different layers are independent, the series of binomials for relative abundance should be replaced by a multinomial to account for the parameter dependencies or the absolute abdundance Poisson model should incorporate overdispersion to model the data more appropriately. Exploiting this room for improvement can lead to further improvement in the performance while the overall framework and usage of MGLEX stay unchanged. When we devised our model, we had an embedding into more complex routines in mind. In the future, the model can be used in inference procedures such as EM or MCMC to infer or improve an existing genome binning. Thus, MGLEX provides a software package for use in other programs. However, it also represents a powerful stand-alone tool for the adept user in its current form.

Currently, MGLEX does not yet have support for multiple processors and only provides the basic functionality presented here. However, training and classification can easily be implemented in parallel because they are expressed as matrix multiplications. The model requires sufficient training data to robustly estimate the submodel weights $\alpha$ using the standard deviation of the empirical log-likelihood distributions and requires linked sequences to estimate $\beta$ using error minimization. Therefore, in situations with a limited number of contigs per genome bin, we advise the generation of linked training sequences of a certain length, as in our simulation; for instance, by splitting assembled contigs. The optimal length for splitting may depend on the overall fragmentation of the metagenome.

Our open-source Python package MGLEX provides a flexible framework for metagenome analysis and binning which we intent to develop further together with the metagenomics research community. It can be used as a library to write new binning applications or to implement custom workflows, for example to supplement existing binning strategies. It can build upon a present metagenome binning by taking assignments to bins as input and deriving likelihoods and $p$-values that allow for critical inspection of the contig assignments. Based on the likelihood, MGLEX can calculate bin similarities to provide insight into the structure of data and community. Finally, genome enrichment of metagenomes can improve the recovery of particular genomes in large datasets.

## ACKNOWLEDGEMENTS

We thank S. Reimering, A. Weimann, and A. Bremges for proofreading and constructive feedback.

### Funding

The authors received no funding for this work.

### Competing Interests

Alice C. McHardy is an Academic Editor for PeerJ.

### Author Contributions

- Johannes Dröge conceived and designed the experiments, performed the experiments, analyzed the data, wrote the paper, prepared figures and/or tables, performed the computation work, and reviewed drafts of the paper.
- Alexander Schönhuth wrote the paper, reviewed drafts of the paper.
- Alice C. McHardy conceived and designed the experiments, wrote the paper, and reviewed drafts of the paper.

### Data Availability

Software:

https://github.com/fungs/mglex/

https://pypi.python.org/pypi/mglex/

Benchmark data and scripts:

Dröge, Johannes (2016). Simulated benchmark metagenome used to demonstrate and evaluate MGLEX software [Data set]. Zenodo. http://doi.org/10.5281/zenodo.201076.

### Supplemental Information

Supplemental information for this article can be found online at http://dx.doi.org/10.7717/peerj-cs.117#supplemental-information.

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
