# Peer review of "A probabilistic model to recover individual genomes from metagenomes"

_PeerJ Computer Science, doi:10.7717/peerj-cs.117_

## Round 0.1 · original submission · Major Revisions

I think the requested changes are not terribly substantial, but there are many of them; the reviewers made some good points and I'd suggest addressing as many of them as possible.

·

Basic reporting

Overall, the manuscript reads well. The introduction gave a good literature review of the problem of recovering genomes from metagenomic data, which led to the probabilistic model proposed. The structure of the manuscript is clear.

Yet I suggest the authors highlight the advantage of the probabilistic model over other methods in both introduction and discussion section. In fact, some of them were shown amid the main text (Line 316-318, 334-338), but a summary of them would be useful to highlight the contribution of this work.

Experimental design

1. Methods section needs polishing

1.a In Line 100, N stands for the number of contigs of a shotgun metagenomic experiment, while in Equation 3,6,8 and 10, N seems to stand for the number of contigs in the genome/bin used to train the parameter θ. A different notation should be used.

1.b The relation between P(contig_i|genome) (line 112) and the likelihood defined by Equation 1 is not clear, especially given that there is not a variable referring to a "genome" in Equation 1. I suggest putting a subscript “g” to the parameter term θ. This could help readers keep in mind that a θ is trained from a genome (or a bin) g, and L(θ_g|F_i) could unambiguously refer to the likelihood of that contig i originated from genome g.

1.c Line 200 and 229 said Naïve Bayes models were chosen. However, the models are not Bayesian (also noted by the authors in line 201). They are simply linear models (after taking logarithm to Equation 7 and 9), and should not be described as “Naïve Bayes models”.

2. Questions for the experimental design

2.a Nielsen et.al (nbt.2939, 2014) found that using the co-abundance across different samples, “species can be segregated accurately using as few as 18 samples”, but not fewer. However, it was noted from line 270 that only four communities of different abundance profiles were simulated in the paper. It is questionable whether four samples are insufficient. BTW, the authors should cite Nielsen et.al’s paper.

2.b The taxonomy annotation was generated from BLAST alignment, according to the supplement. Yet the reference database was not shown. A proper reference database is very crucial in this experiment. To be realistic, the database used for should discard a significant proportion of genomes used to simulate the contigs, otherwise, the BLAST alignments alone would assign most of the contigs to the correct genomes.

Validity of the findings

1. Though it shows that combining the four models resulted in the best MSE and MPC scores, readers are not able to judge how good the aggregate model is. A comparison with other methods (e.g. use Kraken to assign) is welcomed.

2. In the “Genome enrichment” section, there should be another figure showing the % of the target contigs being filtered v.s. p-value cut-off.

3. The statement in line 406 “taxonomic annotations, as used in the model, need not be entirely correct, as long as the same annotation method is applied to all sequences” lacks experimental supports. Already noted in my comment 2.b of "Experimental design", the authors need to select a proper database to support the statement.

Additional comments

The process of creating the simulated dataset was detailed in supplement, but it is still useful to put the datasets accessible online.

The implementation MGLEX is open-sourced on Github and very easy to install via python-pip, which is most appreciated by the community. But it lacks basic document or guidance for users to start with.

·

Basic reporting

my comments are under 'general comments for the author'

Experimental design

my comments are under 'general comments for the author'

Validity of the findings

my comments are under 'general comments for the author'

Additional comments

In their manuscript entitled "a probabilistic model to recover individual genomes from metagenomes" Droege et al describe a model usable to assign contigs to genome bins, as well as evaluating the likelihood that bins are correct. The work is of interest, as genome recovery from metagenomes is only recently becoming a generally used strategy for analysis of metagenomic data.

To provide context for the review: I am a biologist by training, with some experience in analysis of metagenomic data in both a gene and genome centered manner. Most of my comments will thus concern these aspects of the work. I have some remarks on the manuscript, and some on the python implementation of the model.

On the manuscript:

The manuscript describes the model well and is readable for someone who is not intimately familiar with modeling. I was wondering if the implementation of relative abundance as it is described in the manuscript would be sensitive to datasets of uneven sequencing depth. It seems that without standardization between datasets, datasets with more reads wouyld be valued higher, because the variations between genomes in more deeply sequenced datasets is bigger?

I was a little surprised to see that the authors choose to simulate a realistic metagenome, but subsequently choose to avoid a realistic treatment of this dataset. While I see this is valuable in order to assess the performance of the model in a controlled manner, I would like to ask the authors to also show a usecase for their model in which some of the challenges of a real dataset are present. Especially a condition where the training data is not a priori known to be correct (as in a binning of a metagenome assembly) is of interest and I urge the authors to apply their model to either a real dataset, or a de novo assembled simulated dataset in which the available prior knowledge is not used to train the model.

specific comments:
line 28: I think the term binning might have been coined by Banfield and colleagues in their 2004 paper. It would be appropriate to cite that here.
Tyson, Gene W., et al. "Community structure and metabolism through reconstruction of microbial genomes from the environment." Nature 428.6978 (2004): 37-43.

line 80: The authors state that taxonomic affiliation has rarely been used as input in binning. Albertsen et al (2013) do use taxonomic affiliation in their (mostly manual) binning procedure and should be mentioned here. The banfield lab's ggKbase can also use taxonomic info, but I think is not completely open to use and I don't know if there is a paper to cite.

The explanation of using both absolute and relative abundance independently in line 185-189 is not entirely clear to me.

line 228-229 the implementation of taxonomic assignments for contigs is clear, but it is unclear to me what 'level-specific relative frequencies' means for the entries for genomes.

line 253: what is ment by 'amount of contig data'? Total number of bases in contigs or total nr of contigs, or something else?

line 305: missing r in strains

line 305: One of the problems with metagenome analysis is the partial coassembly of (the core genome of) strains, in which case genome abundance is not necessarily capable of distinguishing between strains.

On the python implementation:

Except implicitly on the https://pypi.python.org/pypi/MGLEX website, it is not mentioned that this is a python 3 package, which thwarted my intial attempt at installing it on my macbook

installing the module on our lab server with an anaconda3 environment was straightforward, with the only exception of a permission issue adding lines to easy-install.pth
This might seem oddly specific for a comment, but i encourage the authors to provide a very brief overview of dependencies and installation instructions.

Once installed, the module seems to run fine and the access to the command help is straightforward. However, a general workflow description is lacking. The various commands are listed, but it isn't immediately obvious in which order the commands should be run.

Going through the commands, it seems that 'buildmatrix' is run to generate the responsibility file that most other commands use. It is however unclear to me what the required input 'identifiers' and 'seeds' for this command are, and what file format they should be.
Then 'train' can be used with this responsibility file, to train the model required for 'classify'. As in 'buildmatrix' the input format for the various data required for the model features is unclear.
I suggest that the authors provide a less minimal description of the package use, such that it is accessible to a wider audience in the bioinformatics community

·

Basic reporting

1. In general, the manuscript was well written, with appropriate structure and comprehensive description of the methods and results.

2. In line 355, the sentence of “one would need could quantify bin similarity by direct comparison of
features such as the k-mer vectors or abundances.” is a little bit confusing.

Experimental design

1. In this manuscript, the authors only use simulated data sets to evaluate the method. It will be useful to show how this method can be applied to some real datasets and how this method performs in a more realistic setting. It may be enough to only show the performance on one application, especially the classification/binning. How does this method further improve the quality of binning results generated by other existing binning tools?

2. With the models built using simulated training set, the best performance of MPC is 0.68 using all the features. How does this performance compare to other existing binning tools? It will be interesting to see how existing binning tools work on the “testing data”.

3. More descriptions about the implementation will be appreciated. For example, does the implementation include functions to generate the k-mer frequency or coverage information? This can give the audience valuable information about how easily this tool can be integrated into their pipeline.

Validity of the findings

1. The method described in this manuscript does need a “training dataset” to build the models. Does this training dataset of sequences with known bins have to be generated by other metagenome binning tools? Is it possible to build the models with simulated data as shown in this manuscript? If this method requires some kind of existing sequence bins to work, this method is different from many existing binning tools, which can do the binning from scratch. And what if the performance of existing binning tools on the data set is not good enough? How will the quality of the bins used as training affect the performance of this MGLEX method? It will be appreciated if the authors can make this more clear to the audience and discuss more about how to integrate this method with other existing binning methods to achieve better results.

2. Line 290, “The smaller fraction was used for training because identifying the training data often represents a limiting factor in metagenome analysis”. What does the “limiting factor” mean here? It will be nice to explain the smaller fraction used for training in more details.

3. In line 416, the authors advise the contigs should be splitted to have a certain length. It seems that this is not discussed in other sections of this manuscript. It will be nice to give more explanations about this, like how to determine the proper length.

---

## Round 0.2 · Minor Revisions

A few minor issues have been raised, but I am happy to accept once you address them - no re-review required. Thank you for your efforts!

·

Basic reporting

No comments.

Experimental design

No comments.

Validity of the findings

No comments.

Additional comments

The authors have addressed all my comments.

·

Basic reporting

-

Experimental design

-

Validity of the findings

-

Additional comments

In my opinion the authors have adequately addressed the my comments and those of the other reviewers, and thus recommend the manuscript for publication. I do urge the authors to indeed further expand the documentation.

two nitty gritty remarks:
line 374: euclidian rather than euklidian

line 436-438 has been changed to state that four different submodels are used, but then list coverage, composition and taxonomic asssignment. Maybe explicitly name relative & absolute coverage

·

Basic reporting

The authors made good efforts to address the issues brought up by the reviewers. The authors tested the performance of two other methods, NBC and Centrifuge for comparison. The authors also tested the method on the dataset from caimi-challenge.org, which is helpful. Now I think that all of my comments on the manuscript have been answered.

I just have a question about the discussion about the new Table 3.

Experimental design

no comment

Validity of the findings

On Line 422-423, the authors mention “Table 3 shows that MGLEX bin refinement improved the genome bins in terms of the ARI for both sets of genome bins and  increased the recall for Metawatt but not MaxBin. “ However from Table3, the recalls for Metawatt and Maxbin are both increased to 1.00 for “all contig” , and do not change for “swapped contigs”. Also the ARI for MaxBin “all contigs” decreased from 0.90 to 0.76. This may require further clarification.

---

## Round 0.3 · accepted · Accept

Your manuscript is now ready for publication. Thank you!